# Gait Disorder Detection and Classification Method Using Inertia Measurement Unit for Augmented Feedback Training in Wearable Devices

**DOI:** 10.3390/s21227676

**Published:** 2021-11-18

**Authors:** Hyeonjong Kim, Ji-Won Kim, Junghyuk Ko

**Affiliations:** 1Division of Mechanical Engineering, (National) Korea Maritime and Ocean University, Busan 49112, Korea; koriente@g.kmou.ac.kr; 2Division of Biomedical Engineering, Konkuk University, Chungju 27478, Korea; 3BK21 Plus Research Institute of Biomedical Engineering, Konkuk University, Seoul 05029, Korea

**Keywords:** Parkinson’s disease, gait disorder, augmented feedback training, gait detection, gait classification, wearable device

## Abstract

Parkinson’s disease (PD) is a common neurodegenerative disease, one of the symptoms of which is a gait disorder, which decreases gait speed and cadence. Recently, augmented feedback training has been considered to achieve effective physical rehabilitation. Therefore, we have devised a numerical modeling process and algorithm for gait detection and classification (GDC) that actively utilizes augmented feedback training. The numerical model converted each joint angle into a magnitude of acceleration (MoA) and a Z-axis angular velocity (ZAV) parameter. Subsequently, we confirmed the validity of both the GDC numerical modeling and algorithm. As a result, a higher gait detection and classification rate (GDCR) could be observed at a higher gait speed and lower acceleration threshold (AT) and gyroscopic threshold (GT). However, the pattern of the GDCR was ambiguous if the patient was affected by a gait disorder compared to a normal user. To utilize the relationships between the GDCR, AT, GT, and gait speed, we controlled the GDCR by using AT and GT as inputs, which we found to be a reasonable methodology. Moreover, the GDC algorithm could distinguish between normal people and people who suffered from gait disorders. Consequently, the GDC method could be used for rehabilitation and gait evaluation.

## 1. Introduction

Parkinson’s disease (PD) is one of the most common neurodegenerative diseases among the elderly. The patients’ neural systems become abnormal because the volume of dopamine generated from the nerve cells of the substantia nigra in the brainstem declines. As a result, patients suffer from motor symptoms—such as bradykinesia, tremors, freezing of gait, and non-motor symptoms—such as insomnia and depression. These symptoms diminish the quality of life (QOL) of patients and caregivers. Researchers are studying various methods to treat these symptoms; however, a complete cure for PD has not yet been reported [1,2,3].

Medical staff—such as the doctor in charge—usually prescribe medication and rehabilitation to patients with PD, with rehabilitation being considered a supportive role to medication [4]. However, recent research [5,6,7,8] has provided evidence that rehabilitation can relieve motor and non-motor symptoms, improving a patient’s QOL. Consequently, rehabilitation has recently been considered a primary and independent PD treatment method—that is not simply a supportive treatment method—which can prevent neural degradation [9,10].

Augmented feedback training is a rehabilitation method that enhances cognitive engagement to stop or hold [11,12] patients’ movements, such as maintaining their balance, gait, or tremors. For instance, augmented feedback training attempts to stop patients’ unstable movements by generating feedback to alert patients when stable or ideal movements are observed after irregular actions (which patients usually perform) [11]. Moreover, augmented feedback training attempts to hold stable movements by generating feedback if once unstable actions are observed [12]. The effectiveness of augmented feedback training using these “stop and hand” methods has been clarified in several studies. They showed that training could enhance the ability to maintain balance [13,14,15,16,17,18,19,20,21], gait speed, stride length [22,23,24,25,26], and QOL [27].

However, many researchers are currently studying wearable devices using augmented feedback training or related rehabilitation methods to relieve motor symptoms and maximize the treatment effects of neurodegenerative diseases, particularly PD [28]. For example, Byl et al. [29] researched Smart Shoes to evaluate and observe the effectiveness of visual kinematic feedback on gait training by using Inertia Measurement Unit (IMU) and pressure sensor. Van den Heuvel et al. [23] researched the feasibility of a visual feedback-based balance training system (wearable) and compared its effect with conventional training. Ginis et al. [16] researched the feasibility of the CuPiD system, which is used for effective wearable and portable IMU systems and compared training effects with conventional home-based gait rehabilitation. Carpinella et al. [20] researched the effect on balance and gait outcome as a result of a wearable biofeedback system using a Gamepad System (customized software, PC, and IMU). These researchers showed that wearable devices and their methods were effective on gait factors or tests for gait measurements such as gait speed, FOG-Q, UPDRSIII, BBS, XMWT, FES-I, etc.

While other studies on wearable devices are ongoing, we believe that acceleration and angular velocity near the ankle joint might be useful for detecting and classifying gait. Consequently, we devised a new device for augmented feedback training and its algorithm to create criteria for detecting changes in the magnitude of acceleration (MoA) and Z-axis angular velocity (ZAV) around the ankle joint, classifying the detection results using an exclusive algorithm.

Furthermore, we believe that the MoA and ZAV could be calculated from the hip and knee joint angles. In addition, the hip and knee joint angle patterns significantly affect the MoA and ZAV patterns, although the MoA and ZAV may also be affected by various unknown factors in real life. In this regard, we had to devise a numerical model that could convert joint angles into the MoA and ZAV. The main purpose of the numerical modeling is to observe whether the joint angles are significant factors for gait detection and classification (GDC). Moreover, we had to develop a numerical model that effectively imitated the human gait to overcome certain constraints. The ideal way to verify the capabilities of the designed rehabilitation device was to conduct an experiment with as many participants as possible, since experiments on humans always require a degree of integrity and feasibility as prerequisites.

Consequently, we developed a numerical model to indirectly secure myriad samples. We then developed a GDC algorithm to evaluate the MoA and ZAV. The GDC algorithm was necessary to evaluate the patients’ gait and establish standards to provide feedback based on the MoA and ZAV generated from it. After developing the numerical modeling and GDC algorithm, we evaluated their feasibility based on the results of other studies. Finally, we investigated the feasibility of numerical modeling and the algorithm with experimental participants to obtain GDC results, comparing the experimental results with the numerical modeling results.

## 2. Materials and Methods

### 2.1. Numerical Modeling

#### 2.1.1. Joint Angle Converting

The devices—attached to the ankle joints of users—were designed to analyze their gait. The devices had to able to detect and classify a user’s gait using the acceleration and angular velocity near the ankle joints. Consequently, we had to observe the acceleration and angular velocity trends in the vicinity (when gait was considered to be a reference for device design).

Accordingly, we attempted to estimate the trajectory of the ankle and knee joints during a gait by using the hip and knee joint angles on the sagittal plane, as well as the thigh and shin lengths. We set the definitions of the hip, knee, and ankle joints for Equations (1)–(3), and the estimated joint trajectory trends by changing the joint angle, as follows:(1)$$Hip\;joint\;\{\begin{array}{c} X_{H.n}=1.0\;\\ Y_{H.n}=1.0 \end{array},\;\left(n=1,\;2,\;3\;\cdots 99,\;100\right)\%\;of\;the\;gait\;cycle;$$
(2)$$Knee\;joint\;\{\begin{array}{c} X_{K.n}=X_{H.n}+L1\cdot sin\theta_{H.fl.n}\;\\ Y_{K.n}=Y_{H.n}-L1\cdot cos\theta_{H.fl.n} \end{array},$$
(3)$$Ankle\;joint\;\{\begin{array}{c} X_{A.n}=X_{K.n}+L2\cdot sin(\theta_{H.fl.n}-\theta_{K.fl.n})\;\\ Y_{A.n}=Y_{K.n}-L2\cdot cos(\theta_{H.fl.n}-\theta_{K.fl.n}) \end{array}.$$

Equation (1) represents the fixed position of the hip joint which is used to estimate the position of the knee and ankle joints. Equation (2) represents the estimated knee joint position using the position of the hip joint, the hip joint angle, and the thigh length. Equation (3) represents the estimated ankle joint position using the position of the knee joint, the knee joint angle, and the shin length. Figure 1B shows the representative joint trajectories as a result of the above estimations

Estimating the MoA trend near the ankle joint uses only the ankle joint trajectory with the coefficients listed in Table 1, Equations (4) and (5). Calculating *T* in Table 1 requires the calculation of *D* and *C*. However, the calculation of *D* and *C* is not required if the reference provides the gait stride time. Similarly, the calculation of *D* does not need the calculation of *T* if the reference provides the gait cadence. We used the coefficient $\left(S_{\mathrm{MoA}}\right)$
to convert the raw acceleration data to the sensor value, MoA.

Equation (4) represents the conversion of the ankle joint position to the ankle joint velocity. Equation (5) converts the ankle joint velocity to the ankle joint acceleration. Equation (6) then converts the ankle joint acceleration to the MoA. Figure 2 shows the representative MoA based on these calculations:(4)$$\{\begin{array}{c} {\dot{X}}_{A.n}=\frac{X_{A.n}-X_{A.n-1}}{T}\\ {\dot{Y}}_{A.n}=\frac{Y_{A.n}-Y_{A.n-1}}{T} \end{array},$$
(5)$$\{\begin{array}{c} {\ddot{X}}_{A.n}=\frac{{\dot{X}}_{A.n}-{\dot{X}}_{A.n-1}}{T}\\ {\ddot{Y}}_{A.n}=\frac{{\dot{Y}}_{A.n}-{\dot{Y}}_{A.n-1}}{T} \end{array},$$
(6)$$MoA=S_{MoA}\cdot \sqrt{{\ddot{X}}_{A.n}{}^{2}+{\ddot{Y}}_{A.n}{}^{2}}.$$

Estimating the ZAV trends near the ankle joint during a gait can be achieved using Equations (7)–(9). This requires both the knee and ankle joint trajectories to calculate the direction of a vector from the ankle joint to the knee joint. Equation (7) defines $\vec{V_{n}}$ to be the vector from the ankle joint to the knee joint. Equation (8) defines the vector along the X-axis. The angle between the vector $\vec{V_{n}}$ and the X-axis is calculated using a reverse calculation of the dot product, as expressed in Equation (9). After estimating the angular velocity of the vector, as shown in Equation (10), the ZAV can be derived by multiplying the sensor value by the angular velocity, as expressed in Equation (11). Figure 2 shows that the representative ZAV can be estimated using Equation (11):(7)$$\vec{V_{n}}=\left[\begin{array}{c} {\vec{V}}_{x.n}\\ {\vec{V}}_{y.n} \end{array}\right]=\left[\begin{array}{c} \left(X_{K.n}-X_{A.n}\right)\cdot \hat{i}\\ \left(Y_{K.n}-Y_{A.n}\right)\cdot \hat{j} \end{array}\right],$$
(8)$$\vec{X}=\left[\begin{array}{c} \hat{i}\\ \hat{j} \end{array}\right]\;\left(\;\hat{j}=0\;\right),$$
(9)$$\theta_{yaw.n}={cos}^{-1}\frac{\vec{V_{n}}\cdot \vec{X_{n}}}{\left|\vec{V_{n}}\right|\left|\vec{X_{n}}\right|},$$
(10)$${\dot{\theta }}_{yaw.n}=\frac{\theta_{yaw.n}-\theta_{yaw.n-1}}{T},$$
(11)$$ZAV=S_{ZAV}\cdot {\dot{\theta }}_{yaw.n}.$$

#### 2.1.2. Evaluation Algorithm

Initially, we estimated the MoA and ZAV near the ankle joint using the changes in joint angles during a gait. We then needed to devise GDC methods to use the calculated MoA and ZAV. Consequently, we found significant signs and means could be used for gait detection based on the MoA and ZAV. Figure 2 shows the pattern of MoA and ZAV during one gait cycle.

One significant peak in the ZAV trend was considered to be a strong signal of the user’s motion as it revealed changes in the knee joint angle. Moreover, we assumed that the gait speed and magnitude of the peak might be proportional. Consequently, based on this supposition, we believed we could use the magnitude of the ZAV peak as a first trigger to evaluate the gait and gyroscopic threshold (GT).

Two significant peaks in the MoA trend were considered to be important signals of motion as they revealed that the magnitude of the force was largest when the user walked. In addition, this was observed when the ZAV peak was observed. Consequently, we believed we could use the two MoA peaks as a second trigger for evaluating the gait and acceleration threshold (AT). Accordingly, we devised a GDC algorithm that could reflect these ZAV and MoA triggers. The process can be described as follows:

1.Once the ZAV is recorded to be over the GT, start recording the average MoA while the ZAV is above the GT.2.Confirm the average MoA if the ZAV is recorded to be beneath the GT.3.Then, compare the recorded average MoA with the AT. Classify and detect the result as qualified if the average MoA recorded is over the AT.

Figure 3 shows a flowchart of this process. In the numerical modeling section we devised a numerical modeling method that converted each joint angle, thigh length, and shin length into a MoA and ZAV. Moreover, we devised an algorithm that detected and classified the calculated MoA and ZAV into a qualified gait.

Figure 3 shows how GDC systems work. The initial state is ‘Average MoA = 0’ and ‘N = 0’. Once ‘ZAV’ is recorded over ‘GT’, ‘Average MoA’ begin to be recorded. After ‘ZAV’ is recorded under ‘GT’, the algorithm stops recording ‘Average MoA’. If derived ‘Average MoA’ is bigger than ‘AT’, classify it as ‘Good Gait’, generate feedback, and back to initial state. If derived ‘Average MoA’ is smaller than ‘AT’, back to initial state. 

### 2.2. Gait Characteristic Checking with Numerical Modeling

We planned experiments to indirectly predict the user experience using numerical modeling and confirm the feasibility of the algorithm. To this end, we conducted experiments using reference data from other researchers.

#### 2.2.1. Joint Angle Converting Feasibility Test

We needed to confirm the integrity and feasibility of the process of converting the joint angle into MoA and ZAV. Accordingly, we examined whether the results exhibited a trend similar to using the motion capture system (MCS) (Motion Analysis Corporation, Santa Rosa) and IMU, including acceleration and gyroscopic measurements simultaneously. Consequently, we converted the joint angles from the MCS into MoA and ZAV data and collected MoA and ZAV data using the IMU. Next, we compared the MCS and IMU results in terms of their similarity. The experimental participants performed a gait on a straight path of the MCS 50 times with undecided gait speed and cadence. The participants’ details are shown in Table 2.

#### 2.2.2. Gait Speed Effect Test

The gait speed effect test was conducted to confirm whether gait speed could change GDC results when people who did not have gait disorders performed a gait at different speeds. We also examined how GDC results changed with gait speed, AT, and GT. In this experiment, we utilized recently published data [30]. We confirmed the gait detection and classification rate (GDCR) with 1000 regenerated samples of MoA and ZAV, which followed a normal distribution generated from the reference data.

#### 2.2.3. Gait Disorder Effect Test

We conducted an experiment to observe the effect of a gait disorder on the GDCR, calculated from numerical modeling. In this experiment, we utilized published research data [31] showing the differences in hip and knee joint angles of ordinary people and patients with gait disorders. We confirmed the GDCR with 1000 regenerated samples of MoA and ZAV, which followed a normal distribution generated from the reference data.

#### 2.2.4. Actual GDC Test

We conducted actual experiments to compare the similarities between the results from numerical modeling and experimental results. The experiment participants attached an IMU just above the ankle bone and performed a gait at the decided gait speed and cadence. Gait speeds were fixed at 1.0 km/h, 1.5 km/h, 2.0 km/h, 2.5 km/h, and 3.0 km/h. Cadences were fixed at 90 steps/minute, regardless of gait speed. We collected 180 gait samples for each gait speed. The details of the participants are shown in Table 3. The Hoehn and Yahr Scale [32] was used to evaluate the degree of PD.

All of the patients who participated were diagnosed by medical specialists using history taking, MRI, and survey. One of PD patient (H and Y scale 3) suffered from symptoms as follows: motor symptoms—tremor, bradykinesia (slowdown of movement), and facial expression disorder (hard to express emotion on face). Non motor symptoms—language disorder, depression, and sleep disorder. Another PD patient (H and Y scale 2) suffered from: motor symptoms—bradykinesia, and facial expression disorder. Non motor symptoms—language disorder, depression, sleep disorder, and constipation.

## 3. Results

### 3.1. Joint Angle Converting

The MCS in Figure 4 converted data from the joint angles into MoA and ZAV using numerical modeling. The IMU in Figure 4 is the result of the MoA and ZAV data collected from the IMU—Figure 4A showing the MoA and Figure 4B showing the ZAV. Estimations of the MoA and ZAV from the joint angles was not significantly different from the MoA and ZAV directly collected using the IMU. Consequently, the use of numerical modeling was found to be a reasonable method for estimating gait.

### 3.2. Gait Speed Effect Test

Figure 5 shows the experimental results for evaluating numerical modeling and observing the impact of gait speed. In the figure, we could observe changes in the GDCR by GT, AT, and gait speed. We could observe a pattern showing that a higher GDCR is found at higher speeds if the AT and GT are the same. Conversely, a lower GDCR is found at a higher AT and GT. Although it does not seem to significantly impact the overall trend, we found that several of the evaluations by numerical modeling resulted in a higher GDCR than 1.

### 3.3. Gait Disorder Effect Test

Figure 6 shows the experimental results for evaluating numerical modeling and observing the impact of a gait disorder. We could observe a pattern showing that a lower GDCR is found if the user is affected by a gait disorder. The GDCR could be reduced by a gait disorder at over 15,000 GT and 2000 AT. Similar to the gait speed effect experiments, an increase in the GT and AT reduced the GDCR. Consequently, a gait disorder could be considered to significantly affect (lower) the GDCR.

### 3.4. Actual GDC Test

The results of filtering the MoA and ZAV that occurred during the actual gait using the AT and GT showed that the GDCR was similar to that of the gait speed effect test results (Section 3.2), as shown in Figure 7. Moreover, the user experience of normal and PD patients was similar to that of the gait disorder effect test results (Section 3.3). Overall, a lower GDCR was observed under a higher GT and AT. Conversely, a higher GDCR was observed under a higher gait speed. However, the pattern of the GDCR in PD patients with gait disorders was ambiguous. The GDCR graph surfaces were not evenly separated by gait speed, GT, and AT, even though we requested as comfortable a gait as possible.

## 4. Discussion

In this work, we analyzed the effect of gait speed and gait disorder on the GDCR by converting MoA and ZAV using numerical modeling. We then compared the results with actual experimental results, confirming numerical modeling, the algorithm, and the GDC method to be feasible. In terms of the gait speed effect experiment, the focus was on observing how gait speed affected the GDCR under the same AT and GT conditions: it was confirmed that an increase in the gait speed had an effect on the GDCR, increasing it.

The results of numerical modeling showed that the AT with a GDCR reaching zero varied with gait speed. However, the decrease in the GDCR due to an increase in AT was generally observed to converge close to zero in the vicinity of a specific AT in real situations. Differences in these trends could be attributed to errors due to the following factors: (i) differences in the age of the participants and (ii) differences in the experimental control methods.

First, the participants of reference research, to observe the gait speed effect, were young people around the age of 25. Participants of experiments, to observe the actual experimental results, were middle-aged people around the age of 60. This age difference could lead to a difference in the forces generated during a gait. In the case of young people, they would likely generate more force than the elderly—thus, they could generate a higher MoA. Moreover, they could increase the AT, making the GDCR zero. However, it could also be more difficult for elders to generate a higher MoA that satisfied a specific AT as they might find it difficult to gain sufficient strength to reach the target speeds.

It could also be attributed to differences in the experimental control methods. For example, in the study cited to observe the gait speed effect, the gait speed was controlled by suggesting a specific cadence when walking a fixed distance. In the experiment to observe the actual experimental results, the gait speed was controlled using a treadmill, and a metronome controlled the cadence. The differences in these control conditions may have affected the MoA patterns.

The experimental results to confirm the gait disorder effect showed similar tendencies to the experimental results to confirm the actual results of normal people and patients. In patients with a gait disorder, a low GDCR was observed even at a low AT and GT, both in the numerical modeling and actual experimental results. These results showed that algorithms using changes in the acceleration and angular velocity to determine the MoA and ZAV occurring in a gait in PD patients with a gait disorder, were insufficient compared to using those occurring in normal people.

Other researchers continue research to use IMU as gait analysis and gait rehabilitation. Other researchers have these unique features as follows:

Zhang et al. [33] researched to detect gait patterns with an IMU sensor and air tube connected to pressure sensor. Han et al. [34] researched to observe gait pattern to assist the evaluation of gait. Additionally, Zhao et al. [35] suggested that they could give an examination of gait disorder by detecting the gait cycle event with angular velocity near the heel. Also, Zhou et al. [36] conducted research regarding the measurement of gait speed, stride length, etc., using IMU. Hori et al. [37] suggested a gait estimation method for the gait disorder clinic using IMU.

There were many great pieces of research to help with gait disorder rehabilitation, clinic, monitoring, and evaluating. There may be great studies that we have not identified, however, research interests of the researchers we found were as follows: (i) detection and classification of gait cycle events (heel strike—toe off—heel strike) and its usage for rehabilitation and (ii) gait monitoring or gait evaluation by collecting several factors from gait (gait speed, stride length, stride width, etc.,) using IMU.

This research was a little different from those trends; however, it can use and offer novel factors as follows: (i) concentrating at certain parts of the gait pattern (swing phase), detecting gait by collecting swing speed (ZAV), power of swing (MoA), and using certain filters (AT and GT), and (ii) controlling GDCR by changing certain filters (AT and GT). The GDC method can observe, evaluate, and diagnose gait of users.

To sum it up, we researched about the process of converting acceleration and angular velocity into final data, such as GDCR, using IMU. Moreover, we compared how those factors were shown differently in normal people and PD patients. These themes could not be found in other research and could be meaningful because of the integration of augmented feedback, training, and gait analysis.

The method devised in this paper can be used as follows: (i) rehabilitation of gait disorder, and (ii) diagnosis or evaluation of gait disorder for PD or related disease such as cerebral infarction (stroke), cerebral palsy, and neural related.

The method or a device using the method can be used for rehabilitation of gait disorders that derives from neurological disorders. Patients suffering from gait disorders might not perform a good quality of walking. Therefore, the expected GDCR might be low compared to the GDCR of normal people. 

By lowering AT and GT, patients can generate appropriate GDCR at certain AT and GT, approximately 0.7~0.8 (70~80%). At the AT and GT condition, patients can do rehabilitation to increase their GDCR by receiving augmented feedback. After, the GDCR may be increased if gait symptoms are relieved as a result of rehabilitation. However, the GDCR may not be changed, or may even be decreased, if their gait performance did not improve.

By increasing AT and GT, the method can give a higher goal and lower the GDCR they recorded. They should perform more qualified gait, fast and powerful walking, under the increased AT and GT conditions. The GDC method can relieve gait symptoms with this process, as shown in Figure 8.

The GDC method can also be used for the evaluation and diagnosis of how patients are suffering from gait disorders. The GDC method can return GDCR differently by evaluating the movements of the user’s lower limbs. If patients have severe gait disorder, they will return an irregular and weird pattern of GDCR. However, if a person who has a normal walk with the device using the GDC method, they will return a regular and clear pattern of GDCR. By this method, the GDC method can evaluate and diagnose a user’s walking, as shown in Figure 9.

## 5. Conclusions

We developed a device to determine whether the current gait could be qualitatively satisfied using changes in the MoA and ZAV that occurred near the ankle joint during a gait, and to proceed with augmented feedback training by generating a signal that the gait had been recognized and classified if it were qualitatively satisfied. Moreover, we developed a numerical model that could convert the joint angles into a MoA and ZAV, and developed an algorithm that could evaluate a gait disorder based on the premise that the pattern of PD patients who suffer from a gait disorder is different from that of normal people. As a method to prove this idea, we conducted experiments with normal people and PD patients who were affected by a gait disorder using an IMU and a gait evaluation algorithm. We evaluated the validity of the development by comparing the results with the results of the numerical modeling evaluation.

We confirmed that the designed device could control the GDCR by regulating the AT and GT through the results by applying the algorithm to numerical modeling and the experimental participants’ data. In addition, we checked whether the GDCR could be changed by reflecting the gait speed and gait pattern of the PD patient. Accordingly, the use of the MoA and ZAV, designed through numerical modeling, may be used as representative values of gait patterns that occur near the ankle joint. It was concluded that the use of the algorithm could distinguish whether the device user could produce a normal gait. Moreover, in some cases the algorithm could provide target gait speeds through the adjustment of the GDCR.

The designed device and algorithm could be used as follows: (i) it could induce a gait at a slightly higher speed than the user’s usual gait speed through controlling the AT and GT and informing the user that the current condition is good by generating feedback when the gait speed had reached the target speed. After going through this process several times, gait rehabilitation would allow users to achieve a stronger and faster gait than before; and (ii) after applying the algorithm to the MoA and ZAV of the user’s gait results, it could be used for a gait disorder evaluation and gait pattern detection to observe how different the current user’s gait was compared to normal people through the distribution of the GDCR based on the AT and GT.

## Figures and Tables

**Figure 1 sensors-21-07676-f001:**
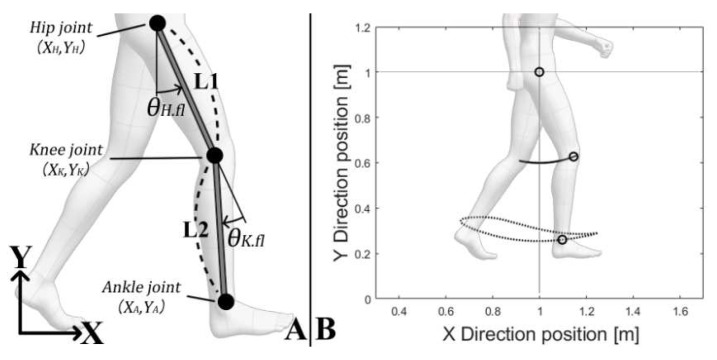
Joint angles converted into joint trajectories.

**Figure 2 sensors-21-07676-f002:**
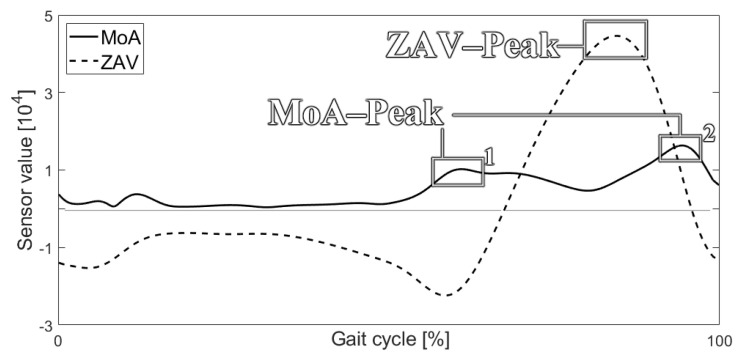
Sample of the converted MoA and ZAV.

**Figure 3 sensors-21-07676-f003:**
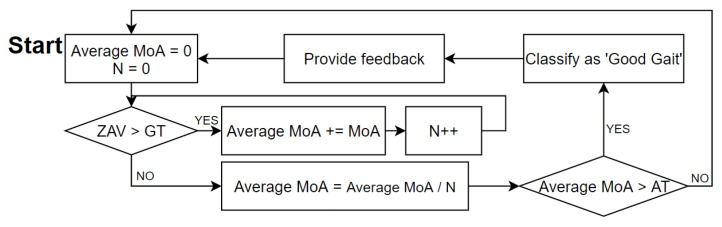
Flowchart explaining the algorithm process.

**Figure 4 sensors-21-07676-f004:**
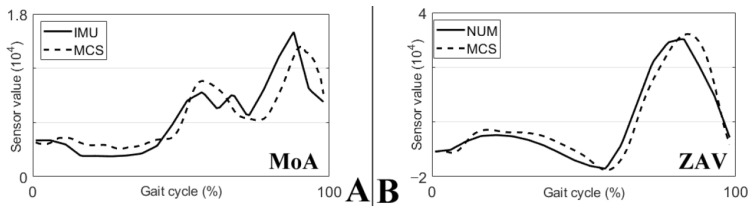
Experimental results of joint angle converting. (**A**) Plotted MoA of the IMU and MCS. (**B**) Plotted ZAV of the IMU and MCS.

**Figure 5 sensors-21-07676-f005:**
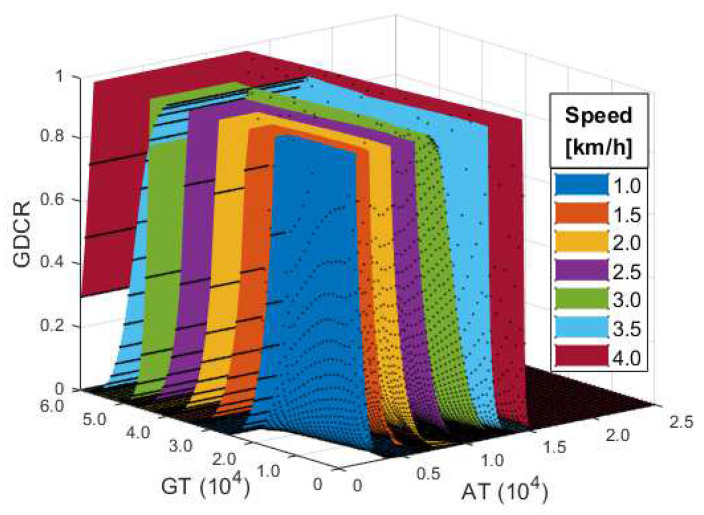
Experimental results of the gait speed effect test.

**Figure 6 sensors-21-07676-f006:**
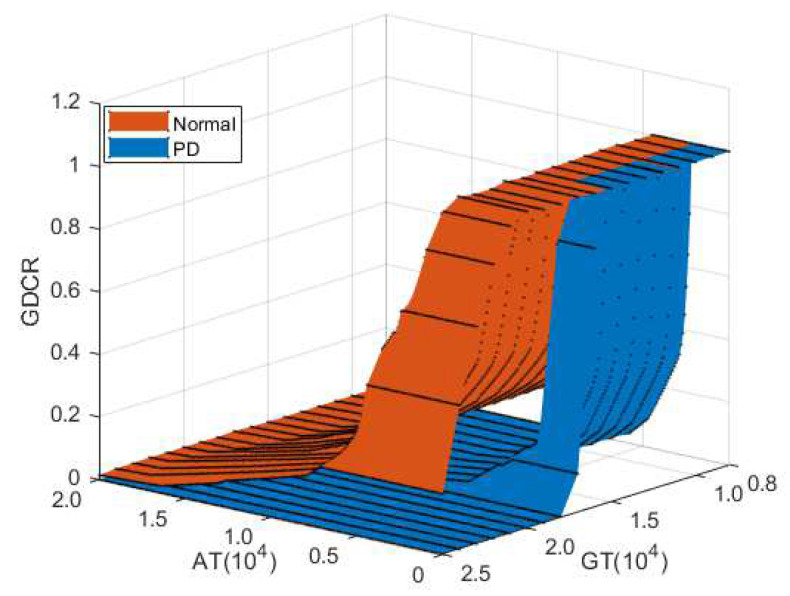
Experimental results of the gait disorder effect test.

**Figure 7 sensors-21-07676-f007:**
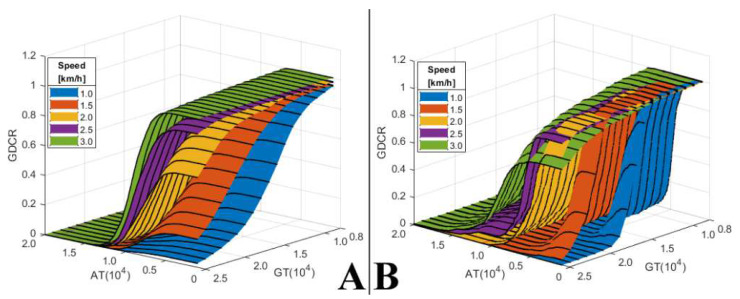
Experimental results of actual GDC test. (**A**) The GDCR of the Non-PD group. (**B**) The GDCR of the PD group.

**Figure 8 sensors-21-07676-f008:**
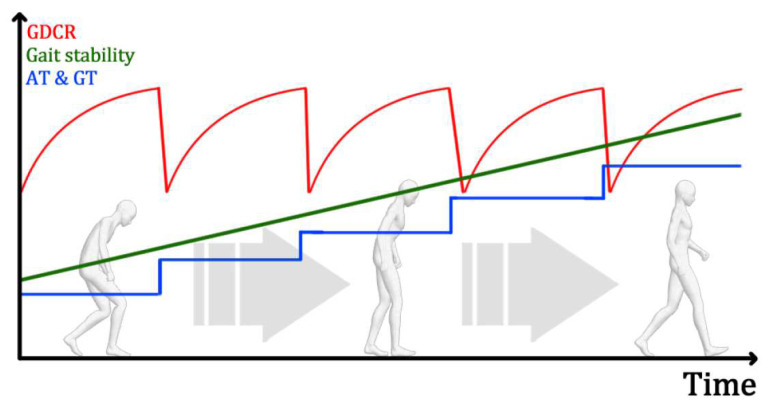
Application of the GDC method. The GDC method could be used for rehabilitation.

**Figure 9 sensors-21-07676-f009:**
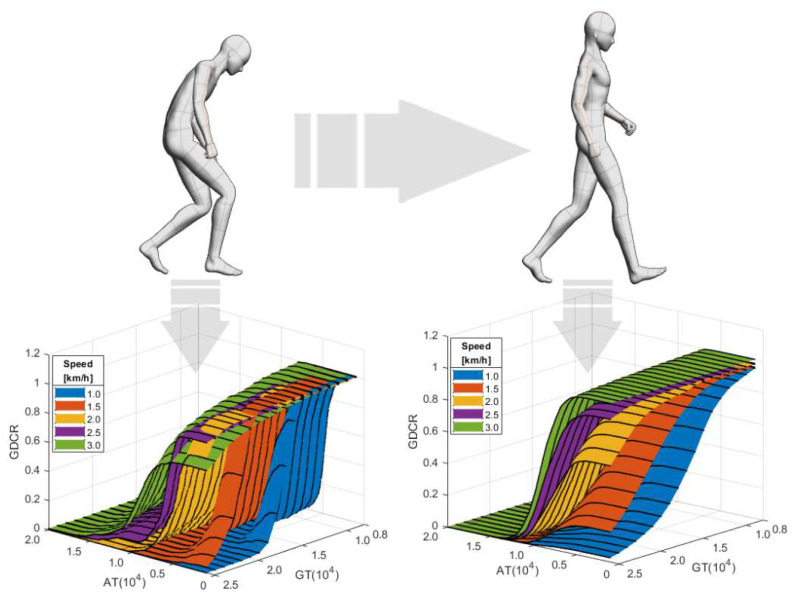
Application of the GDC method. The GDC method could be used for gait disorder evaluation and diagnosis.

**Table 1 sensors-21-07676-t001:** Coefficients used in acceleration and angular velocity calculations.

Coefficient	Mean	Value [Unit]
V	Gait speed	[km/h]
D	Stride length	Max $X_{A.n}$ -Min $X_{A.n}$ [m/step]
C	Cadence during gait	(16.6 V)/D [steps/min]
T	The time required to increase one percent of gait cycle	0.6 C [s/step]
$S_{\mathrm{MoA}}$	Acceleration sensor value sensitivity (model: MPU-6050)	835.07 [LSB·s^2^/m]
$S_{\mathrm{ZAV}}$	Gyroscope sensor value sensitivity(model: MPU-6050)	131 [LSB·s/degree]

**Table 2 sensors-21-07676-t002:** Details of study participants.

Participants	Age	Sex	Height [cm]	Weight [kg]
1	23	Male	171	71
2	22	Male	171	114

**Table 3 sensors-21-07676-t003:** Details of study participants for actual GDC test.

Subject Group	Number of Participants [Number of Females]	Average Age	Average H and Y Scale
Non-PD [NOR]	26 [24]	62 ± 6	Non-PD
PD Patients [PD}	2 [0]	72 ± 0	2.5 ± 0.5

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
