# Peer review of "Gait Disorder Detection and Classification Method Using Inertia Measurement Unit for Augmented Feedback Training in Wearable Devices"

_sensors, 2021, doi:10.3390/s21227676_

Round 1
Reviewer 1 Report
Please find my review comments in teh attached pdf document

Author Response
Dear Whom It May Concern:
Please find enclosed our revised manuscript titled “Gait Disorder Detection and Classification Method using Inertia Measurement Unit for Augmented Feedback Training in Wearable Devices” for publication in Sensors. We thank the review for their detailed comments for improvement. As suggested by the comments, we have made substantial revisions to our manuscript to further strengthen this paper, making it suitable for publication. Please find our responses to the below for further details. The reviewer comments are bolded with our responses give in italics.
Reviewer 1
Comment 1 - Line 97 – 102: The authors represent the human lower limb at the 2D plane as a 2 DOF kinematic structure. Taking into account that a simplified kinematic structure of the lower limb consists of at least 7 DOF (see Olinski et al. 2016, DOI: 10.1515/ijame-2016-0037) at the 3D plane and during walking the ankle movements take place in the 3D plane (especially for walking abnormalities), the authors should discuss the degree of generalization of the approach presented. The authors should also discuss if the inclusion of a more complex kinematic model of the lower limb would result into results that are more accurate and based more in real life walking motion scenarios.
Consideration 1 : Why we used 2 DOF kinematic structure for the GDC method?
The paper (Olinski et al.) mentioned that the joint angle movements are not simple ‘hinge’ motion but results from the complex motion of several muscles. And the authors suggested method to reproduce motion of knee joint (especially, flexion/extension). The main purpose of the research was to devise a numerical model for knee joint motion imitation devices, such as an exoskeleton. An exoskeleton is needed to be designed sophisticatedly because failure to imitate the motion might occur major accidents when people use them. However, the author of paper mentioned that “The movement of the knee joint involves abduction/adduction, as well as internal/external rotation, but it can be simplified to 1 DOF (one degree of freedom) with the main movement of flexion/extension in the body’s sagittal plane” (Olinski et al., Line 2-4). Also the mentioned “the possibility of considering the knee joint only approximately as a hinge is acknowledged in some cases in biomechanics (Huston [8][1]). Moreover, the amount of publications and research in this field like for example (Gerber and Matter [6][2]; Wiczkowski and Skiba [9][3]; Nägerl et al [10][4]) proves that this difference in the motion sometimes is considered very little and is omitted, but it is something that should not be forgotten not only in prosthetic applications, but also when dealing with devices closely cooperating with the human knee joint during movement.” (Olinski et al., Line 35-40) The researchers cited in the paper of Olinski et al. researched about motion observation process of the inside of the knee joint, ‘in vivo’ research in medical terms.
Before generalization and justification ‘why we used numerical modeling based on only 2 DOF?’, the application range of the GDC method should be observed. i) The GDC method can be applied to wearable devices for gait disorder rehabilitation, other ‘in Vitro’ devices, and etc. ii) Motion capture system which counts backward the GDC method. (by converting MoA and ZAV into joint angle) These applications are less dangerous than the example which Olinski et al. and Huston et al. took as a limited example. Therefore, we thought that numerical modeling could be simplified into 1 DOF. As a result, we used 1 DOF at each hip and knee joint.
Consideration 2: Why we used 2D plane only? (why only sagittal plane, not with Frontal and Horizontal?)
In this research, Extension/flexion joint angles on the sagittal plane were used to estimate each joint position. Joint positions can be estimated with the joint angles on the 2D plane, however, additional factors such as adduction/abduction joint angle are needed to estimate joint position on 3D space. This is because we thought that the most important factor to detect gait was the flexion/extension angle resulting from lower limb’s movements, and this study was conducted in an attempt to easily detect gait using a little number of factors as much as possible.
We expected that the GDC method can be more elaborated, for example, evaluating balance during the gait cycle, if adduction/abduction joint angles were considered. The GDC method using IMU will be reinforced after several future works. We will research to make more elaborate GDC method.
Comment 2 - Table 3: The number of PD patients is very low. In parallel, based on the average H&Y score of these two participants, either one of them is at a severe stage of PD (score ~5) and the other one is at a very early stage of PD (score ~0.5) which means that – in accordance with the H&Y scale, either he/she has no walking disorder, or both of them suffer from a slight impairment of balance (score~2.5). It is critical for the particular investigation and the integrity of the results that more PD patients are included. In case that this is not possible, then the two PD patients must be properly characterized and described as the H&Y score affects walking capabilities - (the authors could consult Chatzaki et al. 2021, https://doi.org/10.3390/s21082821).
Thank you for your valuable comment. The reason was Covid-19 broke in early 2020. As you know, conducting experiments with PD patients in the period was quite dangerous because of the possibility of infection. Although the results of two patients who participated in the pre-experiment were used, it was confirmed that the trend was significantly different from that of normal people. In the future, we plan to recruit new experimental participants for additional research. Related contents were added to characterize the degree of PD by listing their symptoms.
Line 209 - 215
--- Revised contents ---
All of the patients who participated were diagnosed by medical specialists using history taking, MRI, and survey. One of PD patient (H&Y scale 3) had suffering from symptoms as follow: Motor symptoms – tremor, bradykinesia (slowdown of movement), and facial expression disorder (hard to express emotion on face). Non motor symptoms – language disorder, depression, and sleep disorder. Another PD patient (H&Y scale 2) had suffering from: Motor symptoms – bradykinesia, and facial expression disorder. Non motor symptoms – language disorder, depression, sleep disorder, and constipation.
--- Revised contents END ---
Comment 3 - Figure 4: In Fig. 4 (A), I observe two peaks of MoA during 60-70% of the gait cycle for the IMU and one peak for MCS. Is this not significantly different? If indeed it is not, please explain which peak must be chosen and why.
First of all, peak of MoA is not a target of choice. The reason for referring to the term, ‘peak’, was to show where the MoA and ZAV were changed noticeably. MoA is used with ZAV by calculating area (Average MoA) when the GDC method detect gait. Therefore, the numbers of peak of MCS and IMU are not significant. Also, we think the results were affected from noise. Overall trend were not broken.
Comment 4 - Line 279 – 281 :As said during the previous comment: What kind of gait disorder? Has it been observed only in Parkinson's patients? Can it be present during other diseases/conditions that affect walking? In addition, can this be used for a differential diagnosis of Parkinson's disease?
We think it can be used for other neurological disease such as cerebral infarction (stroke) and cerebral palsy which show gait disorder as symptoms. Also, it can be used for a diagnosis of PD. However, it is unclear that the GDC method can be presented to other diseases. (Together with the second comment) Nowadays, we think Covid-19 is getting less dangerous. We are planning to observe more clear possibility and feasibility of the GDC methods with the application of PD patients and other gait disorder patients by collaborating with hospitals. Future work will observe clear possibility and feasibility of the GDC method.
We have addressed the reviewer’s comments by revising our manuscript making it suitable for publication. Thank you for your consideration of our revised manuscript.
Sincerely Yours,
Dr. Junghyuk Ko
Assistant Professor
Department of Mechanical Engineering
Korea Maritime and Ocean University
Email: jko@kmou.ac.kr
Telephone: (82)051-410-4292
[1] Huston R.L. (2008): Principles of Biomechanics. – CRC Press, ISBN: 978-0-8493-3494-8.
[2] Gerber C. and Matter P. (1983): Biomechanical analysis of the knee after rupture of the cruciate ligament and its primary repair. An instant-centre analysis of function. – The J. of Bone and Joint Surgery, vol.65-B, No.4, pp.391-399.
[3] Wiczkowski E. and Skiba K. (2008): Kinetic analysis of the human knee joint. – Biology of Sport, vol.25, No.1, pp.77-91.
[4] Nägerl H., Dathe H., Fiedler CH., Gowers L., Kirsch S., Kubein-Messenburg D., Dumont C. and Wachowski M.M. (2015): The morphology of the articular surfaces of biological knee joints provides essential guidance for the construction of functional knee endoprostheses. – Acta of Bioengineering and Biomechanics, vol.17, No.2, pp.45-53.

Reviewer 2 Report
Summary:
This paper proposes a gait detection algorithm that is targeted to Parkinson's disease. Authors describes the algorithms to detect the gait pattern with making a mathematical model. Evaluations are performed by using experimental data from acquiring gait movement of health participants and published data from advanced research outcomes. They conclude that the proposed algorithm is effective to detect the disorder during gait movement.
Comments:
- In Table 1, it is not understandable that how the S_MOA and S_ZAV are derived. Is it a typical value in the datasheet of inertial sensors?
- There is no explanation for Figure 3. The bock diagram should be an important part of the algorithm. But it is not clear that how the analysis works because the detailed explanation of the figure.
- What is the main implementation of the proposed algorithm? Unit? Device? System? It seems a system because it is implemented by MCS and IMU. The manuscript should be revised to explain it clearly.
- In this paper, the originality is not clearly described. The mathematical model seems intuitively defined. But, there is no comparisons to the other advanced research outcomes. Authors have to discuss related works with disseminating the differences and advantages of the proposed method.
- It is not clear how the experimental evaluations (described in the manuscript as "test"s) are contributing for aid or support for the patients of the Parkinson's disease. The result just discusses detection availabilities. But, how those results contributes to the patients should be discussed.
Author Response
Dear Whom It May Concern:
Please find enclosed our revised manuscript titled “Gait Disorder Detection and Classification Method using Inertia Measurement Unit for Augmented Feedback Training in Wearable Devices” for publication in Sensors. We thank the review for their detailed comments for improvement. As suggested by the comments, we have made substantial revisions to our manuscript to further strengthen this paper, making it suitable for publication. Please find our responses to the below for further details. The reviewer comments are bolded with our responses give in italics.
Reviewer 2
This paper proposes a gait detection algorithm that is targeted to Parkinson's disease. Authors describes the algorithms to detect the gait pattern with making a mathematical model. Evaluations are performed by using experimental data from acquiring gait movement of health participants and published data from advanced research outcomes. They conclude that the proposed algorithm is effective to detect the disorder during gait movement.
Comment 1 - In Table 1, it is not understandable that how the S_MOA and S_ZAV are derived. Is it a typical value in the datasheet of inertial sensors?
Yes. It is typical value of Inertia Measurement Unit (IMU). 835.07, S_MOA which given in paper is sensor value of MPU-6050. Also, 131, S_ZAV is value of same module. It can be differed by model and its setup. For example, S_MOA was derived as follows: We used MPU-6050 as basic setup and its sensitivity of accelerometer was 8192 LSB/g (gravity). It converted to 835.07 Sensitivity of gyroscope was 131 LSB/(°/s).
We revised information about sensor value in table 1. Revised contents were bolded.
--- Revised contents ---
Table 1. Coefficients used in acceleration and angular velocity calculations.
|
Coefficient |
Mean |
Value [Unit] |
|
V |
Gait speed |
[km/h] |
|
D |
Stride length |
Max -Min [m/step] |
|
C |
Cadence during gait |
(16.6 V)/D [steps/min] |
|
T |
The time required to increase one percent of gait cycle |
0.6 C [s/step] |
|
|
Acceleration sensor value sensitivity (model: MPU-6050) |
835.07 [LSB·s2/m] |
|
Gyroscope sensor value sensitivity (model: MPU-6050) |
131 [LSB·s/degree] |
--- Revised contents END ---
Comment 2 - There is no explanation for Figure 3. The bock diagram should be an important part of the algorithm. But it is not clear that how the analysis works because the detailed explanation of the figure?
We added how the algorithm works.
‘Initial state is ‘Average MoA = 0’ and ‘N = 0’. Once ‘ZAV’ recorded over ‘GT’, ‘Average MoA’ begin to be recorded. After ‘ZAV’ recoded under ‘GT’, the algorithm stop recording ‘Average MoA’. If derived ‘Average MoA’ is bigger than ‘AT’, classify it as ‘Good Gait’, generate feedback, and back to initial state. If derived ‘Average MoA’ is smaller than ‘AT’, back to initial state.’
Line 166 - 170.
--- Revised contents ---
Figure 3. Flowchart explaining the algorithm process.
Figure 3 shows that how GDC systems work. Initial state is ‘Average MoA = 0’ and ‘N = 0’. Once ‘ZAV’ recorded over ‘GT’, ‘Average MoA’ begin to be recorded. After ‘ZAV’ recoded under ‘GT’, the algorithm stop recording ‘Average MoA’. If derived ‘Average MoA’ is bigger than ‘AT’, classify it as ‘Good Gait’, generate feedback, and back to initial state. If derived ‘Average MoA’ is smaller than ‘AT’, back to initial state.
--- Revised contents END ---
Comment 3 - What is the main implementation of the proposed algorithm? Unit? Device? System? It seems a system because it is implemented by MCS and IMU. The manuscript should be revised to explain it clearly.
The algorithm is proposing Gait Detection and Classification systems. Therefore, we revised to explain the purpose of algorithm.
Line 166 - 170.
--- Revised contents ---
Figure 3. Flowchart explaining the algorithm process.
Figure 3 shows that how GDC systems work. Initial state is ‘Average MoA = 0’ and ‘N = 0’. Once ‘ZAV’ recorded over ‘GT’, ‘Average MoA’ begin to be recorded. After ‘ZAV’ recoded under ‘GT’, the algorithm stop recording ‘Average MoA’. If derived ‘Average MoA’ is bigger than ‘AT’, classify it as ‘Good Gait’, generate feedback, and back to initial state. If derived ‘Average MoA’ is smaller than ‘AT’, back to initial state.
--- Revised contents END ---
Comment 4 - In this paper, the originality is not clearly described. The mathematical model seems intuitively defined. But there is no comparisons to the other advanced research outcomes. Authors have to discuss related works with disseminating the differences and advantages of the proposed method.
We added related contents in manuscript.
Line 292 - 316.
--- Revised contents ---
Other researchers continue research to use IMU as gait analysis and gait rehabilitation. Other researches have these unique features as follows : Zhang et al.[37] researched to detect gait pattern with IMU sensor and air tube connected to pressure sensor. Han et al.[38] researched to observe gait pattern to assist evaluation of gait. Additionally, Zhao et al.[39] suggested that they could give examination of gait disorder by detecting gait cycle event with angular velocity near heel. Also, Zhou et al.[40] conducted research about measurement of gait speed, stride length, and etc. using IMU. Hori et al.[41] suggested gait estimation method for gait disorder clinic using IMU. There were many great pieces of research to help gait disorder rehabilitation, clinic, monitoring, and evaluating. There may be great studies that we have not identified, however, research interests of researches we found were as follows: i) Detection and classification of gait cycle events ( heel strike – toe off – heel strike) and its usage for rehabilitation. ii) Gait monitoring or gait evaluation by collecting several factors from gait (gait speed, stride length, stride width, and etc.) using IMU.
This research was a little different from those trends, however, can use and offer novel factors as follows: i) Concentrating at part of gait pattern (Swing phase), detecting gait by collecting swing speed (ZAV), power of swing (MoA), and using certain filters (AT and GT). ii) Controlling GDCR by changing certain filters (AT and GT). The GDC method can observe, evaluate, and diagnose gait of users.
To sum it up, we researched about the process of converting acceleration and angular velocity into final data, such as GDCR, using IMU. Moreover, we compared how those factors were shown differently from normal people and PD patients. These themes could not be found in other research and could be meaningful because of the integration of augmented feedback training and gait analysis..
--- Revised contents END ---
Comment 5 - It is not clear how the experimental evaluations (described in the manuscript as "test"s) are contributing for aid or support for the patients of the Parkinson's disease. The result just discusses detection availabilities. But, how those results contributes to the patients should be discussed.
We added related contents and figures in 4. Discussion section.
Line 317 - 344.
--- Revised contents ---
The method devised in this paper can be used for as follows: i) Rehabilitation of gait disorder ii) Diagnosis or evaluation of gait disorder for PD or related disease such as cerebral infarction (stroke), cerebral palsy, and neural related.
- i) The method or a device using the method can be used for rehabilitation of gait disorders that come from neurological disorders. Patients suffering from gait disorders might not perform good quality of walking. Therefore, the expected GDCR might be low compared to the GDCR of normal people.
By lowering AT and GT, patients can generate appropriate GDCR at certain AT and GT, approximately 0.7 ~ 0.8 (70% ~ 80%). At the AT and GT condition, patients will do rehabilitation to increase their GDCR, by receiving augmented feedback. After, the GDCR may be increased if gait symptoms are relieved as a result of rehabilitation. However, the GDCR may not be changed or even be decreased if their gait performance did not improve.
By increasing AT and GT, the method can give a higher goal and lower the GDCR they recorded. They should perform more qualified gait, fast and powerful walking, under the increased AT and GT condition. The GDC method can relieve gait symptoms with this process, as shown in figure 8.
Figure 8. Application of the GDC method. The GDC method could be used for rehabilitation.
- ii) The GDC method can be used for the evaluation and diagnosis of how patients are suffering from gait disorders. The GDC method can return GDCR differently by evaluating the movements of the user’s lower limb. If patients have severe gait disorder, they will return irregular and weird pattern of GDCR. However, if a person who is a normal walk with the device using the GDC method, they will return a regular and clear pattern of GDCR. By this method, the GDC method can evaluate & diagnose user’s walking, as shown in figure. 9
Figure 9. Application of the GDC method. The GDC method could be used for gait disorder evaluation and diagnosis.
--- Revised contents END ---
We have addressed the reviewer’s comments by revising our manuscript making it suitable for publication. Thank you for your consideration of our revised manuscript.
Sincerely Yours,
Dr. Junghyuk Ko
Assistant Professor
Department of Mechanical Engineering
Korea Maritime and Ocean University
Email: jko@kmou.ac.kr
Telephone: (82)051-410-4292

Reviewer 3 Report
- "The devices had to able to "should be "The devices had to be able to " . The participants are all at 22 or 23 years old. Is that appropriate for this study which focus on elderly people having Parkinson’s disease?
- It is suggested that more details should be discussed about wearable devices in introduce but not insert citation. Many readers may not be familiar with this technique.
- In line 131, what are “significant signs and interpretative meanings”?
- Figure.5 shows the relationship between GDCR and gait speed and lower、acceleration threshold (AT) 、 gyroscopic threshold (GT). Both higher speeds, lower AT, and GT are significant to obtain higher GDCR. But the lower bound of AT and GT should be limited to ensure the accuracy of GDCR.
- The advantages of GDC numerical modeling over other methods, should be described in the discussion to reflect the research. Such as Inertial measurement units combined with a smartphone application (CuPiD-system), etc.
- There are several grammar and proofreading errors that need to be corrected throughout the paper. Please go through and carefully correct these.
Author Response
Dear Whom It May Concern:
Please find enclosed our revised manuscript titled “Gait Disorder Detection and Classification Method using Inertia Measurement Unit for Augmented Feedback Training in Wearable Devices” for publication in Sensors. We thank the review for their detailed comments for improvement. As suggested by the comments, we have made substantial revisions to our manuscript to further strengthen this paper, making it suitable for publication. Please find our responses to the below for further details. The reviewer comments are bolded with our responses give in italics.
Reviewer 3
Comment 1 - "The devices had to able to "should be "The devices had to be able to " .The participants are all at 22 or 23 years old. Is that appropriate for this study which focus on elderly people having Parkinson’s disease?
The experiment they participated was conducted to observe the method of converting joint angles to IMU sensor value. Regular and frequent gait were needed to observe similarities between the methods. Therefore, they were needed to the experiment because young participants were expected to perform regular gaits. It is true that we should compare experimental results between normal elders and PD patients. Therefore, we compared their results in ‘Actual GDC test’ experiment
Comment 2 - It is suggested that more details should be discussed about wearable devices in introduce but not insert citation. Many readers may not be familiar with this technique
We revised the introduction section to seem more information about wearable devices and their effect, without additional citation.
Line 57 – 70
--- Revised contents ---
However, many researchers are currently studying wearable devices using augmented feedback training or related rehabilitation methods to relieve motor symptoms and maximize the treatment effects of neurodegenerative diseases, particularly PD [28]. For example, Byl et al. [29] researched Smart Shoes to evaluate and observe the effectiveness of visual kinematic feedback on gait training by using Inertia Measurement Unit (IMU) and pressure sensor. van den Heuvel et al. [30] researched the feasibility of visual feedback-based balance training system (wearable) and compared its effect with conventional training. Ginis et al. [31] researched the feasibility of CuPiD system, which is used for effective wearable and portable IMU systems and compared training effects with conventional home-based gait rehabilitation. Carpinella et al. [32] researched the effect on balance and gait outcome as a result of a wearable biofeedback system using Gamepad System (customized software, PC, and IMU). Those researches showed that wearable devices and their method were effective on gait factors or tests for gait measurements such as gait speed, FOG-Q, UPDRSIII, BBS, XMWT, FES-I, and etc..
--- Revised contents END ---
Comment 3 - In line 131, what are “significant signs and interpretative meanings”?
We wrote that to notice ‘ what kind of signals on MoA and ZAV were important for gait detection ’. We revised the sentence to show why we mentioned like that.
Line 140 – 141
--- Revised contents ---
Consequently, we found significant signs and means could be used for gait detection based on the MoA and ZAV. Figure 2 shows pattern of MoA and ZAV during one gait cycle.
--- Revised contents END ---
Comment 4 - Figure.5 shows the relationship between GDCR and gait speed and lower, acceleration threshold (AT), gyroscopic threshold (GT). Both higher speeds, lower AT, and GT are significant to obtain higher GDCR. But the lower bound of AT and GT should be limited to ensure the accuracy of GDCR
We thought that the meaning of your comments is 'show data of figure 5 to seem more intuitive' to show the relation between AT, GT, and GDCR. Thus, we revised figure 5.
Line 228 - 229
The former figure was version of before revision. The latter figure is revised version of figure 5.
Comment 5 - The advantages of GDC numerical modeling over other methods, should be described in the discussion to reflect the research. Such as Inertial measurement units combined with a smartphone application (CuPiD-system), etc.
We added related contents in manuscript.
Line 292 - 316.
--- Revised contents ---
Other researchers continue research to use IMU as gait analysis and gait rehabilitation. Other researches have these unique features as follows: Zhang et al.[37] researched to detect gait pattern with IMU sensor and air tube connected to pressure sensor. Han et al.[38] researched to observe gait pattern to assist evaluation of gait. Additionally, Zhao et al.[39] suggested that they could give examination of gait disorder by detecting gait cycle event with angular velocity near heel. Also, Zhou et al.[40] conducted research about measurement of gait speed, stride length, and etc. using IMU. Hori et al.[41] suggested gait estimation method for gait disorder clinic using IMU. There were many great pieces of research to help gait disorder rehabilitation, clinic, monitoring, and evaluating. There may be great studies that we have not identified, however, research interests of researches we found were as follows: i) Detection and classification of gait cycle events ( heel strike – toe off – heel strike) and its usage for rehabilitation. ii) Gait monitoring or gait evaluation by collecting several factors from gait (gait speed, stride length, stride width, and etc.) using IMU.
This research was a little different from those trends, however, can use and offer novel factors as follows: i) Concentrating at part of gait pattern (Swing phase), detecting gait by collecting swing speed (ZAV), power of swing (MoA), and using certain filters (AT and GT). ii) Controlling GDCR by changing certain filters (AT and GT). The GDC method can observe, evaluate, and diagnose gait of users.
To sum it up, we researched about the process of converting acceleration and angular velocity into final data, such as GDCR, using IMU. Moreover, we compared how those factors were shown differently from normal people and PD patients. These themes could not be found in other research and could be meaningful because of the integration of augmented feedback training and gait analysis..
--- Revised contents END ---
Comment 6 - There are several grammar and proofreading errors that need to be corrected throughout the paper. Please go through and carefully correct these.
This paper was revised and edited about English language with professional editing service.
We have addressed the reviewer’s comments by revising our manuscript making it suitable for publication. Thank you for your consideration of our revised manuscript.
Sincerely Yours,
Dr. Junghyuk Ko
Assistant Professor
Department of Mechanical Engineering
Korea Maritime and Ocean University
Email: jko@kmou.ac.kr
Telephone: (82)051-410-4292
Round 2
Reviewer 1 Report
Thank you for your response to my previous comments.
Reviewer 2 Report
Authors of this manuscript responded appropriately to my negative comments. However, the English should be revised hardly, especially the revised parts (for example, "researched" should be "investigated"). It would be better to use the English editing service of MDPI or to ask someone to proofread once who is familiar to English writing.